# Novel Molecular Mechanisms Involved in the Medical Treatment of Pulmonary Arterial Hypertension

**DOI:** 10.3390/ijms24044147

**Published:** 2023-02-19

**Authors:** Irene Martin de Miguel, Alejandro Cruz-Utrilla, Eduardo Oliver, Pilar Escribano-Subias

**Affiliations:** 1Pulmonary Hypertension Multidisciplinary Unit, Cardiology Department, Hospital Universitario 12 de Octubre, 28041 Madrid, Spain; 2Instituto de Investigación Sanitaria Hospital 12 de Octubre (i+12), 28041 Madrid, Spain; 3Centro de Investigación Biomédica en Red de Enfermedades Cardiovasculares (CIBERCV), 28029 Madrid, Spain; 4Centro de Investigaciones Biológicas Margarita Salas (CIB-CSIC), 28040 Madrid, Spain

**Keywords:** pulmonary arterial hypertension, vascular remodeling, endothelial dysfunction, BMP signaling, tyrosine kinase receptor, inflammation, epigenetics, mitochondrial dysfunction

## Abstract

Pulmonary arterial hypertension (PAH) is a severe condition with a high mortality rate despite advances in diagnostic and therapeutic strategies. In recent years, significant scientific progress has been made in the understanding of the underlying pathobiological mechanisms. Since current available treatments mainly target pulmonary vasodilation, but lack an effect on the pathological changes that develop in the pulmonary vasculature, there is need to develop novel therapeutic compounds aimed at antagonizing the pulmonary vascular remodeling. This review presents the main molecular mechanisms involved in the pathobiology of PAH, discusses the new molecular compounds currently being developed for the medical treatment of PAH and assesses their potential future role in the therapeutic algorithms of PAH.

## 1. Introduction

Pulmonary arterial hypertension (PAH) represents a rare disorder characterized by pulmonary vascular remodeling, with subsequent increase in pulmonary vascular resistance (PVR), pulmonary artery pressures and right ventricular afterload, and finally right ventricular dysfunction and right heart failure, which represents the leading cause of death in this population [1,2]. PAH is currently defined by a mean pulmonary artery pressure > 20 mmHg, a pulmonary arterial wedge pressure ≤ 15 mmHg, and a PVR > 2 Wood units by right heart catheterization [3]. Despite the significant progress in the understanding of the pathophysiological mechanisms and its natural history, the advances in the medical therapy of PAH, and the improved survival of this population in recent years [4,5], this disease remains incurable and is associated with substantial morbidity and mortality [6,7,8,9,10].

Current treatment strategies are aimed at pulmonary vasodilation to ultimately reduce right ventricular afterload [11,12,13,14,15,16]; however, there is no approved treatment to date directly targeting and antagonizing the remodeling of the pulmonary vasculature [3,17,18]. In this review, the authors present an overview of the new therapeutic targets for the medical treatment of PAH with focus on their molecular mechanisms, analyze the available scientific evidence and the potential future role of these novel molecules in the therapeutic algorithm of PAH.

## 2. Pathobiology of PAH

### 2.1. Pathological Changes in the Pulmonary Circulation and the Right Ventricle

The pathobiology of PAH is complex and comprises vasoconstriction, abnormal cell proliferation, fibrosis, in situ thrombosis, and inflammation of mainly small muscular-type pulmonary arteries (50–500 μm) [2,19]. Although the underlying mechanisms are not completely understood, a variety of factors (hypoxia, shear stress, genetic and epigenetic alterations, drugs and toxins, infections, autoimmune reactions) may induce endothelial dysfunction which initiates subsequent structural and functional changes in the pulmonary vasculature [20]. The dysfunctional endothelium promotes vasoconstriction, depicts reduced anticoagulant properties and an altered metabolism, and has an increased expression and production of certain adhesion molecules and growth factors that will provoke uncontrolled proliferation, impaired angiogenesis and excessive inflammatory and immune activation [1,17,18,21].

Remodeling occurs in the three layers of the pulmonary vessel. Intimal fibrosis and neointima formation results from uncontrolled proliferation and apoptosis of pulmonary artery endothelial cells (PAEC); pulmonary artery smooth muscle cell (PASMC) hyperplasia causes medial hypertrophy; and fibroblast proliferation, collagen disruption and infiltration of inflammatory cells leads to remodeling of the adventitia [2]. Additionally, PASMC migration prompts neomuscularization of previously non-muscularized arterioles, and in situ thrombosis resulting from excessing platelet activation in contact with a dysfunctional endothelium. Dysregulation of the innate and adaptative immune system and presence of a marked pro-inflammatory state further contribute to vascular remodeling [17,22].

All these phenomena lead to progressive luminal narrowing, occlusion and loss of pulmonary vessels. In advanced stages of the disease plexiform lesions develop in the adventitia of the pulmonary vasculature as a consequence of disordered PAEC proliferation and neoangiogenesis, and are thought to correspond to anastomosis between bronchial vessels or vasa vasorum and pulmonary vessels, thus creating intrapulmonary right-to-left shunting between the pulmonary and bronchial (systemic) circulation [2,17,23]. Although PAH was originally thought to predominantly affect small muscular-type pulmonary arteries, more recent work has shown that also arterioles, capillaries and venules develop different degrees of pathological changes in all types of pulmonary hypertension, with venular remodeling not being restricted to patients with pulmonary veno-occlusive disease [17]. Eventually, the increased systemic blood flow into arterioles, capillaries and venules through plexiform lesions could prompt venular and venous remodeling. Characteristic venous and venular pathological changes include intimal fibrosis with partial or complete lumen occlusion, PASMC hyperplasia or capillary dilation and proliferation [24].

Ultimately, the remodeling of the pulmonary circulation falls on the right ventricle. Increased arterial afterload prompts right ventricular hypertrophy. Adaptative concentric hypertrophy with minimal fibrosis and appropriate myocardial microcirculation contrasts with maladaptive eccentric hypertrophy, rarefaction of myocardial microcirculation and significant fibrosis, and whether the former or the latter develops remains poorly understood, although derangements in angiogenesis, adrenergic signaling and metabolism (increased glycolysis and glutaminolysis) seem to be involved [1,21,25,26]. Maladaptive right ventricular hypertrophy is associated with altered ventriculo-arterial coupling, decrease or loss of ventricular contractile reserve, ischemia, higher levels of biomarkers, and worse functional capacity and survival [27]. Finally, right ventricular systolic dysfunction develops which represents the major prognostic determinant of patients with PAH [2,3,4,5,6,7,8,9]. Interestingly, despite the development of systolic dysfunction, right ventricular contractility as measured by end-systolic elastance is usually not decreased or even increased. Nevertheless, etiologic-specific differences have been observed, with systemic sclerosis-associated PAH depicting decreased sarcomeric function as opposed to the hypercontractility from idiopathic PAH [28].

Despite the right cardiac chambers being the main ones affected as a direct consequence of the increased pulmonary arterial afterload, development of left ventricular diastolic dysfunction represents a relevant aspect that might difficult clinical management, especially after lung transplantation. Diastolic dysfunction of the left ventricle results from a chronic state of low left ventricular preload derived from ventricular interdependence (leftward displacement of the interventricular septum with left ventricular compression due right-sided pressure overload and right-left ventricular asynchrony with pathological prolongation of left isovolumic relaxation) and from low right cardiac output [29,30]. Diastolic dysfunction has been shown to increase morbidity and mortality after lung transplantation, and acute pulmonary edema may develop after surgery, since a chronically underfilled left ventricle with impaired relaxation and compliance, and fibrotic remodeling suddenly receives a normal cardiac output from a right ventricle that pumps back into a low-pressure pulmonary circulation [31].

### 2.2. Cellular and Molecular Abnormalities, Genetic and Epigenetic Alterations

In recent years, significant progress has been made in the understanding of the molecular pathways that are thought to be involved in the development of PAH; nevertheless, much still remains unknown. Genetic, epigenetic and environmental factors are thought to cause dysregulation in gene transcription (both aberrant activation and depletion of gene expression) that leads to altered cell function and subsequently to the derangements encountered in the pulmonary vasculature [32,33]. Specifically, it has been suggested that genetic and environmental elements lead to abnormal secretion and function of growth factors, cytokines, hormones and ion channels that activate different signaling pathways which, in turn, regulate transcription factors. The latter also recruit chromatin-modifying enzymes whose role consists in DNA and histone modification, hence, epigenetic phenomena. The transcription factors and chromatin-modifying enzymes, accordingly, regulate the pulmonary vascular transcriptome and activate the production of proliferative, pro-inflammatory and metabolic genes that lead to the remodeling of the pulmonary vasculature [17,34,35].

Understanding the pathobiology of PAH is of key importance to develop treatments targeted at specific molecular regulators that interrupt and reverse the adverse pulmonary vascular remodeling, and promote regeneration of normal pulmonary vessels. The main molecular signaling pathways involved in the pathobiology of PAH are presented below (Figure 1).

#### 2.2.1. Bone Morphogenetic Protein (BMP) Signaling

The bone morphogenetic protein receptor type-2 (BMPR2), a serine/threonine kinase transmembrane receptor, is a member of the transforming growth factor-β (TGFβ) superfamily. This receptor is part of the BMP signaling pathway which is involved in several important cellular processes (cell differentiation, proliferation, apoptosis) in many tissues. Autosomal-dominant mutations in the gene encoding BMPR2 represent the most common mutation encountered in PAH, since they have been documented in 70–80% of hereditary cases of PAH, and in up to 40% of idiopathic PAH; and reduced BMPR2 expression has even been observed in patients with PAH without mutations in BMPR2 [36,37,38]. Nevertheless, mutations in BMPR2 have a reduced penetrance (lifetime risk ~20%), suggesting that additional genetic and/or environmental factors (‘second hits’) may contribute to disease development [39].

In the pulmonary vasculature, BMPR2 has a high expression in the endothelium, and less in the PASMC [40]. BMP signaling occurs through a heterodimeric complex of two type-1 receptors and two type-2 receptors. Specifically to BMPR2, 2 BMPR2 (type-2 receptor) bind the BMP ligand and then form an active complex with two units of the activin-like kinase (ALK) receptor (type-1 receptor). Downstream signaling involves a canonical (Smad-dependent [Smad1/5/8]) and a non-canonical (Smad-independent, through the extracellular signal-regulated kinase, Janus kinase, p38 mitogen-activated protein kinase, phosphatidylinositol 3-kinase, and LIM kinase-1) pathway, and both routes have been involved in PAH. Ultimately, activation of transcription factors and expression of specific genes occurs [41,42]. Further modulation of this pathway takes place through BMP co-receptors (endoglin, betaglycan) and antagonists (gremlin, noggin, chrodin), and signaling inhibitors (Smad 6/7, FK-binding protein-12) [43].

Pathogenic mutations in all the main domains of BMPR2 have been documented, and, although at a lower frequency, mutations in other related genes such as *BMP9*, *BMP10*, *ALK*, *SMAD8/9* or *endoglin* have been observed in some studies, highlighting the importance of this axis for the pulmonary vasculature homeostasis [44,45,46,47]. Dysregulation of the BMP pathway leads to an imbalance between anti-proliferative and pro-proliferative routes (Figure 2A): decreased BMPR2 activity (anti-proliferative effects) causes reduced BMP9/10-mediated signaling, and thus increased apoptosis and reduced endothelial barrier integrity [40]. This promotes the production of activin A, growth differentiation factors (GDF) 8 and 11 that bind to the ALK (type-1 receptor)—activin receptor type IIA (ActRIIA, type-2 receptor) complex (from the TGFβ superfamily) and activate the pro-proliferative Smad2/3 pathway. This route also increases the expression of the BMP antagonists, gremlin and noggin, further downregulating the BMPR2 pathway [48,49,50].

Different studies have been and are being carried out to develop molecular compounds that act on the altered mechanisms resulting from the different mutations in BMPR2: decreased receptor expression, DNA damage, decreased receptor transportation to the cell membrane, imbalance between activin/GDF and BMP signaling, or additional factors (second hits) that further decrease BMPR2 expression [51].

#### 2.2.2. Tyrosine Kinase Receptors

Platelet-derived growth factor (PDGF) is the most potent mitogenic for vascular smooth muscle cells and binds to the tyrosine kinase platelet-derived growth factor receptor (PDGFR), subtypes α and β. These receptors regulate a significant range of biological functions, including cell growth, motility, differentiation or metabolism. Aberrant expression and function of PDGF and of the corresponding receptors (PDGFR), has been documented in PAH, which leads to excessive proliferation of PASMC in the arteriolar media and myofibroblasts in the vessel lumen, with subsequent media hypertrophy and intimal fibrosis (Figure 3A) [52].

Crosstalk between the PDGFR and the TGFβ/BMPR2 pathways has been documented. Detailed analysis of transcriptome and proteome changes induced by PDGF-BB (a PDGFR ligand) in PASMC of rats showed that PDGF-BB-mediated upregulation of miRNA-376b led to decreased expression of BMPR2, which promoted proliferation of PASMC [53]. Additionally, some downstream effectors from the non-canonical routes of TGFβ seem to be shared with the PDGF axis, and crosstalk between the two upregulated pathways in PAH may occur, thus promoting additive mitogenic effects in the pulmonary vasculature [54].

Two additional PDGFR-related kinases have also been involved in the pathobiology of PAH: the colony stimulating factor 1 receptor (CSF1R) and the tyrosine-protein kinase KIT (c-KIT). CSF1R is expressed on monocytes and macrophages, which secrete PDGF ligands and inflammatory cytokines, thereby promoting inflammation and remodeling of the pulmonary vasculature [55,56]. C-KIT is expressed on endothelial progenitor cells and mast cells, and it has been observed in plexiform lesions from patients with PAH. This receptor is thought to play a role in perivascular inflammation and vascular remodeling [57].

The critical role of these tyrosine kinase receptors in the pathobiology of PAH makes them an appealing therapeutic target in PAH.

#### 2.2.3. Potassium Channels

Loss or reduction of expression and function of potassium channels has been shown to be involved in the pathobiology of PAH, and restoration of these channels ameliorated or reversed PAH in experimental models [58,59]. To date, genetic changes in three families of potassium channels have been observed in association with PAH: the *KCNK3* gene that encodes the channel KCNK3 (or TASK-1) that belongs to the two-pore-domain potassium channels (K2P); the *ABCC8* gene that encodes the subunit SUR1 of the ATP-sensitive potassium channel (KATP) that corresponds to the inwardly-rectifying potassium channels (Kir); and the voltage-gated potassium channels (Kv) [60,61,62]. Heterozygous loss-of-function mutations in *KCNK3* were identified in different families of patients with heritable PAH and in patients with idiopathic PAH by whole exome sequencing [63]. Furthermore, expression and function of KCNK3 was significantly reduced in patients with heritable and idiopathic PAH without *KCNK3* mutations, and in rat models of monocrotaline-induced PAH. Various intracellular signaling pathways (signal transducer and activator of transcription 3/nuclear factor of activated T-cells, miRNA) and several growth factors, vasoconstrictors or cytokines involved in PAH including PDGF, BMPR2 or interleukin-17 have been found to modulate KNCK3 channel expression and activity [17,64,65,66]. Loss-of-function of the KCNK3 channel is thought to lead to plasma-membrane depolarization, pulmonary artery vasoconstriction and proteomic alterations that cause proliferation of PAEC, PASMC and adventitial fibroblasts, pulmonary inflammation and increase in right ventricular systolic pressure [58,67]. Heterozygous loss-of-function mutations in *ABCC8* have also been reported in patients with PAH and it may promote decreased apoptosis and enhanced inflammation and hyperplasia, although the involved molecular mechanisms remain unclear [68,69]. Finally, although to date there are no known mutations in Kv channels, reduction of the expression or function of these voltage-gated potassium channels has been observed in PAH, which might lead to dysregulation of cell proliferation and death and promotion of excessive vasoconstriction [62,70,71]. In summary, mutations in the *KCNK3* and *ABCC8* genes may be involved in genetically predisposed PAH, while KCNK3, KATP or Kv channels’ dysfunction may contribute to non-hereditary forms of PAH.

#### 2.2.4. Transcription Factors and Transcriptional Coregulators

The different pathological stimuli (hypoxia, shear stress, oxidative stress, mitogens, inflammation) activate the signaling cascades and downstream transcription factors and transcriptional coregulators. The latter represent proteins that do not bind to the DNA itself but to transcription factors to activate or inhibit them. This stimulus-specific activation of transcription factors and transcriptional coregulators in PAH promotes the expression of pro-proliferative, pro-inflammatory and metabolic genes. Numerous transcription factors and coregulators have been involved in the pathobiology of PAH. Downregulation of anti-proliferative and anti-inflammatory transcription factors such as the peroxisome proliferator-activated receptor gamma (PPAR-γ) in endothelial cells leads to impaired cell survival and angiogenesis [72], and has been observed to be a relevant regulator of the BMP/TGFβ balance [73]; while inactivation of Forkhead box O1 promotes proliferation and decreases apoptosis of PASMC [74]. Aberrant activation of the transcriptional coregulator C-terminal binding protein 1 has been involved in fibroblast proliferation and apoptosis resistance and glycolytic reprogramming [21].

#### 2.2.5. Epigenetics, Metabolic and Mitochondrial Dysfunction

In addition to genetic mutations and dysregulation of transcription factors, altered expression of different elements involved in epigenetic regulation have been documented in PAH, and include abnormal levels of histones, histone deacetylases, miRNA or long-non-coding RNA segments, or aberrant DNA methylation of specific genes [75]. MiRNAs have recently gained importance in the pathobiology of PAH, and at least some seem to play a role in the regulation of the BMP/TGFβ pathway (e.g., miR-140-5p activates BMPR2, while the miR-130/301 family upregulates the pro-proliferative TGFβ route) [73,76].

Furthermore, metabolic derangements with reduced mitochondrial glucose oxidation and upregulation of glycolysis have been observed in the pulmonary vasculature (in PAEC, PASMC, fibroblasts, macrophages) and in myocytes of the right ventricle [33]. The transcription factor PPARγ, which is a downstream target of BMPR2, plays a key role in glucose homeostasis (and vascular remodeling), and its decrease in PAH is thought to be a key element among the factors involved in the metabolic alterations of this disease [73]. Derangements in the fatty acid metabolism have also been observed, with right ventricular steatosis and lipotoxicity from triglyceride metabolites and reactive oxygen species, possibly due to increased fatty acid delivery to the ventricle, fatty acid synthesis, and reduced oxidative mitochondrial capacity [40,77]. Interestingly, mitochondrial products could play a regulatory role of transcription factors and epigenetic mechanisms [34]. Therefore, intervening on the epigenetic-metabolism pathway may represent a potential therapeutic strategy [26].

#### 2.2.6. Immune System and Inflammatory Response

Dysregulation of the inflammatory and immune responses, possibly secondary to environmental and genetic factors, has been demonstrated in experimental models and in tissues from patients with PAH, and plays a role in endothelial dysfunction, vascular remodeling and ventricular dysfunction. Both the innate and adaptative systems are altered, and perivascular infiltration of T and B lymphocytes, dendritic cells, mast cells or macrophages next to pulmonary vessels has been documented, occasionally even before vascular remodeling develops. T-helper 17 cell immune polarization occurs (which has also been documented in chronic inflammatory and autoimmune diseases) while there is impairment of T-regulatory cell function, and circulating autoantibodies can be found in the absence of an autoimmune condition [22]. Furthermore, the various cell types from the pulmonary vasculature (PAEC, PASMC, myofibroblasts, fibroblasts) are in a significant pro-inflammatory state as reflected by the increased expression and secretion of different cytokines and chemokines, and of inflammatory cell adhesion molecules (intercellular adhesion molecule 1, vascular cell adhesion molecule 1, E-selectin). Augmented levels of interleukins-1 and -6, leukotriene B4, leptin receptors or tumor necrosis factor α, or inactivation of the Forkhead box O1 pathway are involved in adverse pulmonary vascular changes via different signaling pathways [74,78,79,80,81,82].

Furthermore, there seems to be bidirectional interaction between the BMP signaling axis and the immune system [83]. Reduced BMPR2 expression promotes macrophage activation through the granulocyte-macrophage colony stimulating factor [84]. Endothelial macrophage migration leads to endothelial cell adhesion and further recruitment of macrophages and lymphocytes. BMPR2-dependent release of interleukin-6 from PASMC under stress conditions could act as additional paracrine activator of macrophages, thereby perpetuating the inflammatory state. These accumulated macrophages secrete, among others, high levels of the enzyme leukotriene A4 hydrolase which produces leukotriene B4 that has been involved in PAEC apoptosis and proliferation and hypertrophy of PASMC [83,85].

## 3. New Molecular Compounds Targeting the Altered Pathways

Treatment strategies available to date are aimed at antagonizing the inappropriate vasoconstriction that occurs in PAH and target three different major signaling pathways: (1) the nitric oxide and soluble guanylate cyclase pathway (phosphodiesterase inhibitors [sildenafil, tadalafil] and soluble guanylate cyclase stimulators [riociguat]); (2) the endothelin pathway (endothelin receptor antagonists [ambrisentan, bosentan, macitentan]) [12,13]; and (3) the prostacyclin pathway (prostacyclin analogues [epoprostenol, iloprost, treprostinil] and prostacyclin receptor agonist [selexipag]) [3,11,14,86,87]. Current and upcoming preclinical and clinical research is focused on the development of compounds that directly act in the different molecular pathways associated with the development of PAH to counteract the obstructive vascular remodeling and small vessel loss [56,88,89]. Specifically, drugs targeting various pathogenic mechanisms are being developed: bone morphogenetic protein signaling, tyrosine kinase receptors, serotonin metabolism, angiogenesis, extracellular matrix, estrogens or epigenetics [51,90,91]. Significant scientific advances in preclinical research including new molecular tools, omics technologies, human tissue biobanks and in vivo models, have served to better understand the pathobiology and molecular mechanisms involved in PAH, to develop and test new molecular compounds as adjunctive treatment of PAH that can be subsequently validated in clinical trials (translational research).

Next, the different developing compounds according to each molecular pathway are discussed.

### 3.1. Reduction of the Inappropriate Proliferation

#### 3.1.1. BMP Signaling

Since the BMP signaling axis is a major element in the pathobiology of PAH, developing molecular compounds that modulate this pathway represents an attractive therapeutic target. Sotatercept is a first-in-class fusion protein that is made of the extracellular domain of the ActRIIA attached to the Fc domain of human IgG1. This molecule acts as a ligand trap by binding activins and GDFs and thus restoring the balance between pro-proliferative and anti-proliferative BMP pathways (Figure 2B). It is administered as subcutaneous injection every 21 weeks. Sotatercept represents by far the drug that has accumulated the largest evidence as potential future treatment for PAH. It has been investigated in phase-2 (PULSAR, PULSAR-open label extension, SPECTRA) and phase-3 trials (STELLAR), and has demonstrated efficacy and a good safety profile [88,89]. Regarding tolerability and safety of this compound, the most common adverse events are thrombocytopenia, increased hemoglobin levels, telangiectasis, headache, diarrhea and nasopharyngitis. The pathophysiological mechanisms behind thrombocytopenia remain ununderstood, while the elevation in hemoglobin levels is dose-dependent due to its erythropoietic effects. PULSAR was a randomized, double-blind, placebo-controlled trial where patients with PAH in WHO functional class II or III were randomized to sotatercept (dose of 0.3 mg or 0.7 mg per kg of body weight) or placebo in addition to background therapy (35% double, 56% triple therapy, 37% prostacyclin infusion, with no significant differences between groups). Compared to placebo, sotatercept significantly reduced the PVR (primary endpoint) in a dose-dependent fashion, and, interestingly, this was driven by a decrease in mean pulmonary artery pressure, with minimal change in cardiac output and pulmonary arterial wedge pressure, suggesting that this compound might exert a direct effect on pulmonary vascular remodeling, in contrast to the main vasodilatory effect of other compounds. The secondary endpoints of 6-min walk distance (6MWD), N-terminal pro-B-type natriuretic peptide (NT-proBNP) levels, functional class, and clinical worsening also favored the sotatercept group. The drug was well tolerated in the majority of patients, with thrombocytopenia and an increased hemoglobin levels being the most frequent hematologic adverse events [88]. The long-term efficacy and safety of sotatercept were evaluated in the open-label extension of the PULSAR trial at months 18–24; 92% of the participants continued into this period. There were significant improvements in the primary and secondary endpoints (PVR, 6MWD, NT-proBNP, functional class, mean pulmonary artery pressure, right atrial pressure), with no differences in cardiac output and pulmonary arterial wedge pressure; proving long-term safety and efficacy of sotatercept in PAH. Fewer treatment-emergent adverse events occurred with longer exposure to the drug, and incidence of these were similar with both doses of sotatercept, supporting the choice of the 0.7 mg/kg dose for the subsequent studies. Serious adverse events were reported in 4.8% of the participants, but none led to drug discontinuation. None of the deaths (2.9%) were considered drug-related [89]. Additionally, preliminary results from the concomitant SPECTRA study (NCT03738150), a single-arm, open-label, exploratory study assessing the effect of sotatercept on top of background therapy by invasive cardiopulmonary exercise testing, revealed improvements in several functional and ventilatory metrics and pulmonary pressures [92]. Subsequently, the STELLAR trial (NCT04576988), the phase-3, randomized, double-blind, placebo-controlled study further assessing safety and efficacy of sotatercept (dose 0.7 mg/kg) on top of background PAH therapy at 24-weeks was carried out and the results were recently released in press. The primary endpoint of increase in the 6MWD was met for sotatercept. Eight of nine secondary endpoints were also achieved: multicomponent improvement (6MWD, NT-proBNP, functional class), PVR, NT-proBNP, functional class amelioration, time to death or clinical worsening, achievement of a low-risk status, improvement of physical and cognitive/emotional scores. The overall safety profile was consistent with what had been observed in the phase-2 trial [93].

The potential therapeutic role of BMP9 has also been explored in experimental studies, suggesting that treatment of BMPR2-deficient PAEC with BMP9 ameliorated the endothelial barrier and decreased apoptosis; and in rodent experimental PAH models BMP9 increased BMPR2 expression and downstream signaling, reversing the PAH phenotype [94].

An additional compound of potential therapeutic interest is tacrolimus, a calcineurin inhibitor currently used for immunosuppression of patients with solid-organ transplants. It has been shown to both activate BMP signaling (via the canonical Smad-dependent pathway) and antagonize the inhibition of the BMPR2 by the FK-binding protein-12 [95]. Preclinical and preliminary clinical studies reported favorable effects of tacrolimus of reversal of vascular remodeling and improvement of symptoms, exercise capacity, 6MWD and NT-proBNP [96]. A more recent, phase-2a randomized, placebo-controlled trial (TransformPAH) showed no statistically significant differences in the 6MWD, NT-proBNP levels, or echocardiographic right ventricular function parameters. Nevertheless, the small sample size of the study and the heterogeneity of the response could have limited its statistical power, and sub-analyses suggested that in some patients there was a correlation between BMPR2 expression and right ventricular function. Large multicenter studies are needed to establish whether tacrolimus may represent a potential therapeutic compound in PAH [97].

#### 3.1.2. Tyrosine Kinase Receptors

The central role of aberrant tyrosine kinase receptor signaling in PAH justifies the development of compounds targeting these pathways as treatment of PAH (Figure 3B).

Imatinib, a tyrosine kinase receptor inhibitor (PDGFR α and β, DDR, c-KIT, CSF1R, ABL) approved for the treatment of chronic myeloid leukemia, had proven to reverse PAH in animal models and suggested to be clinically and hemodynamically effective as adjunctive treatment of patients with PAH in case reports and a phase-2 trial [98,99,100]. Subsequently, the IMPRES trial, a phase-3, randomized, double-blind, placebo-controlled trial was designed and conducted, which was followed by an open-label extension study (up to 204 weeks). Compared to placebo, imatinib led to a statistically significant and clinically meaningful increase in the 6MWD and improvement of hemodynamics. Nevertheless, safety concerns arose, due to common serious adverse events and drug discontinuation, including the development of subdural hematoma in eight patients receiving imatinib and anticoagulation [101,102]. These safety concerns and common drug intolerance have precluded to date the use of oral imatinib as treatment of PAH. To avoid the systemic adverse effects and drug intolerance, the safety and efficacy of a new formulation of this compound, namely dry powder inhaled imatinib, is currently being investigated in a phase-2b/3 (IMPAHCT, NCT05036135; IMPAHCT-FUL, NCT05557942) clinical trial (Figure 3C,D) [103,104].

Furthermore, seralutinib, a highly potent (13- to 20-fold greater than imatinib) and specific inhaled tyrosine kinase inhibitor that targets PDGFR, CSF1R and c-KIT, has been specifically designed as a treatment for PAH (Figure 3C,D). Preclinical studies involving seralutinib have reported efficacy in preventing PAH progression and vascular remodeling, restored lung BMPR2 expression, reduced levels of pro-inflammatory biomarkers, improved hemodynamics, decreased NT-proBNP levels, and showed greater efficacy when compared with imatinib [56]. A phase-2, randomized, double-blind, placebo-controlled trial, the TORREY study, evaluating the safety and efficacy of inhaled seralutinib has recently announced its results: the primary endpoint (decrease in PVR), as well as several secondary endpoints (6MWD, NT-proBNP, echocardiographic structural and functional parameters) were met for seralutinib. Additionally, the compound was acceptably well tolerated, with less common and less severe adverse events and no report of subdural hematoma [90].

#### 3.1.3. Angiogenesis and Endothelial Function

Endothelial dysfunction with reduced nitric oxide synthesis by the endothelial nitric oxide synthase (eNOS) is central in the pathobiology of PAH. Therefore, the autologous administration of patient-derived endothelial progenitor cells transfected with the *eNOS* gene has been investigated. The election of endothelial progenitor cells as vehicle for the delivery of *eNOS* lies in the fact that this type of cells are bone marrow-derived cells that favor endothelial hemostasis and angiogenesis after an insult, and they have rendered positive results in experimental models of PAH [105]. A subsequent phase-1 dose escalation study (PHACeT) of this type of cell therapy was conducted in seven patients, with no sustained hemodynamic improvements, but a significant increase in the 6MWD. A serious adverse event (sudden death) occurred in one patient with end-stage disease immediately after discharge from the study procedure [106]. Currently, a phase-2, randomized, double-blind, placebo-controlled trial (SAPPHIRE, NCT03001414) is being conducted to further assess safety and efficacy of endothelial progenitor cells-eNOS [107].

### 3.2. Serotonin Metabolism

High levels of serotonin have been documented in patients with PAH. Furthermore, appetite suppressants like fenfluramine and dexfenfluramine, which increase serotonin release from platelets are established causes of drug-associated PAH [3]. Serotonin is synthesized from the amino acid L-tryptophan through the enzyme tryptophan hydroxylase. Genetic overexpression, augmented uptake in the cytoplasm of PASMC and increased plasmatic levels of serotonin induced PAH and vascular remodeling in experimental models, whereas tryptophan hydroxylase deficiency or inhibition prevented from the development of hypoxia-induced PAH in rodents [108]. Accordingly, clinical studies have been developed to assess the role of antagonizing serotonin metabolism as potential treatment in PAH. Currently, the ELEVATE2 study (NCT04712669), a phase-2b, randomized, double-blind, placebo-controlled trial is ongoing and will address the effect of rodatristat ethyl, a peripherally restricted inhibitor of the enzyme peripheral tryptophan hydroxylase, on PVR in patients with PAH [109].

### 3.3. Inflammation, Oxidative Stress and Immune Modulation

The chronically altered inflammation and immune regulation encountered in PAH have led to the development of numerous studies aimed at analyzing the role of anti-inflammatory and immune modulatory agents as treatment of PAH [81,82,83]. While preclinical models have shown promising results of different compounds, clinical studies have rendered heterogenous results.

Rituximab, a widely used anti-CD20 antibody that depletes B-cells expressing CD20, was investigated in a phase-2, randomized, double-blind, placebo-controlled trial in patients with systemic sclerosis-associated PAH. The drug was well tolerated but it failed to meet the primary endpoint of change in the 6MWD [110]. Tocilizumab, an interleukin-6 receptor neutralizing antibody, was assessed in a phase-2 open-label trial (TRANSFORM-UK), where it did not show any change in PVR or in other secondary efficacy parameters compared to placebo. Subgroup analysis suggested that patients with connective tissue disease could have a modest decrease in PVR, although this should be interpreted with caution [111]. Anakinra is an interleukin-1 receptor antagonist that was tested in a phase-1b/2, open-label, single-arm pilot study of patients with PAH where it showed to reduce inflammation and improve heart failure symptoms, but larger studies to confirm these results are warranted [112]. Finally, other compounds like the tumor necrosis factor α inhibitor etanercept has shown beneficial effects in preclinical models of PAH, but has yet not been tested in clinical trials [51].

Bardoxolone methyl represents a compound that promotes antioxidant gene transcription, increase in ATP and oxidative phosphorylation via the activation of the Nrf2 signaling pathway and suppression of NF-kB transcriptional activity. It was investigated in the LARIAT study (NCT02036970), a phase-2, randomized, double-blind, placebo-controlled trial, and interim analyses revealed higher 6MWD in the bardoxolone methyl group compared to placebo [113]. The two phase-3 studies that followed (CATALYST, NCT02657356; RANGER, NCT03068130) were terminated prematurely due to an estimated excessive risk for the PAH patients to be exposed to in-person visits during the SARS-CoV-2 pandemic [114,115].

### 3.4. Estrogen Signaling

The epidemiological differences in the prevalence of PAH with a 4:1 ratio for women has caused great interest in establishing the role of estrogens and derived metabolites in the pathobiology of PAH, and consequently in assessing the potential therapeutic role of estrogen inhibitors in this disease. Experimental models have shown that estrogen signaling decreases BMPR2 expression and promotes PASMC proliferation through estrogen receptors [116]. Nevertheless, despite women having a higher prevalence of PAH, male sex is a predictor of worse prognosis, which is thought to be related to beneficial effects of estrogen on the right ventricle (the so-called ‘estrogen paradox’), rendering the potential therapeutic use of compounds targeting estrogen signaling complicated.

Anastrozole is an inhibitor of aromatase, the enzyme that catalyzes the formation of estradiol from testosterone. This compound was investigated in a small, randomized, double-blind, placebo-controlled pilot study where it decreased the 17b-estradiol levels (E2) but had no effect on the tricuspid annular plane systolic excursion (co-primary endpoint) or echocardiographic metrics; the 6MWD was significantly increased, although it did not reach the clinically meaningful established cut-off of 33 m [117]. A multicenter double-blind, placebo-controlled, phase-2 randomized clinical trial (PHANTOM, NCT03229499) has been subsequently conducted to further assess safety and efficacy of this compound in PAH (results unpublished) [118]. As second approach, the efficacy and safety of the estrogen receptor antagonist tamoxifen is currently being evaluated in a phase-2 randomized, double-blind, placebo-controlled trial (T3PAH, NCT03528902) [119].

### 3.5. DNA Damage and Repair

Chronic exposure of endothelial cells to shear stress, reactive oxygen species, inflammation and excessive immune activation leads to DNA damage in lung vascular cells. Failure in DNA damage repair by the endothelial cells due to chronic exposure to insults results in accumulation of mutations and chromosomal rearrangements [120]. Additionally, crosstalk between BMPR2 signaling and DNA damage sensors has been observed [51]. Accordingly, interest has been gained in developing compounds aimed at reversing DNA damage. Poly (ADP-ribose) polymerase-1 (PARP1) usually contributes to DNA repair but it can also act under specific circumstances as tumorigenic. PARP1 is overexpressed in PAH, and Veliparib, a PARP1 inhibitor, reversed PAH in experimental models [120]. This has led to the development of a phase-1b clinical trial (OPTION, NCT03782818) testing Olaparib, a selective PARP1 inhibitor, to assess the safety of this compound in patients with PAH [121].

### 3.6. Epigenetics

Bromodomain-containing protein 4 regulates the transcription of genes involved in the pathobiology of PAH through epigenetic phenomena by binding acetylated histone tails, and it has been observed to be significantly elevated in PASMC and lung tissue from patients with PAH. Inhibition of bromodomain-containing protein 4 by oral apabetalone has been shown to reverse pulmonary vascular remodeling in preclinical studies [122]. The APPRoAcH-p trial, an open-label, single-arm, pilot study aimed at assessing the safety and hemodynamic effects of oral apabetalone in patients with PAH, revealed a good tolerability of the compound, as well as a potential therapeutic benefit (decrease in PVR and improvement in cardiac function), that needs further confirmation in future studies [123].

### 3.7. Right Ventricle Neurohormonal and Metabolic Modulators

Similar to the neurohormonal blockade pursued in left heart failure, various preclinical and clinical studies have been conducted to analyze the effect of these compounds on right heart failure that develops in PAH, with mainly negative results and potentially dangerous effects such as decrease in blood pressure or in heart rate. Therefore, the use of angiotensin-converting enzyme inhibitors, angiotensin receptor blockers, angiotensin receptor–neprilysin inhibitor, sodium–glucose cotransporter-2 inhibitors, beta-blockers, or ivabradine is not recommended by the current European guidelines for the diagnosis and treatment of pulmonary hypertension, unless required for comorbidities [3]. Spironolactone has shown positive results in animal models and is currently being tested in a randomized, placebo-controlled trial (STAR-HF, NCT03344159) [124].

Lastly, targeting the metabolic derangements that take place in PAH has also been pursued. Ranolazine is thought to act on glucose and fatty acid metabolism, although its mechanism of action on the right ventricle is not completely understood. It improved right ventricular function, left ventricular end-diastolic volumes and biventricular stroke volumes compared to placebo in a small randomized trial [125].

## 4. Discussion and Future Directions

Despite the significant advances in the management of patients with PAH, survival is still significantly reduced, hence the need to redefine the therapeutic algorithms to include compounds targeting the underlying pathobiological mechanisms of PAH. To date, the modulation of the BMPR2 signaling pathway by the molecular compound sotatercept represents a realistic fourth pathway to the current three classical therapeutic pathways for PAH. It remains unknown, however, how sotatercept will be positioned in the current therapeutic algorithm. Whether other compounds currently under investigation will be further incorporated to the therapeutic arsenal for PAH remains to be elucidated. The new inhaled formulation of the tyrosine kinase receptor inhibitors imatinib and seralutinib may represent another promising therapeutic alternative, and question remains, as whether both the BMPR2 and the tyrosine kinase pathways will be combined together as adjunctive treatments to the current three pathways. Furthermore, the inhaled administration of imatinib and seralutinib aims to maintain their therapeutic benefit without the systemic adverse effects of oral administration. Although it is evident that inflammation and the immune system play an important role in the pathobiology of PAH, the clinical studies conducted to date have failed to demonstrate a therapeutic benefit of compounds targeting the altered immune regulation. Eventually, the complexity of the immune system and the numerous interactions between the different inflammatory pathways, immune cells and regulatory elements, render it very difficult for a single compound directed against a specific target to exert a net therapeutic benefit. Additionally, the degree of contribution of inflammation and the immune system to the remodeling of the pulmonary vasculature might be highly variable in each patient. Finally, genetics and epigenetics, although they currently are in a more initial stage of research development, constitute a field that will very likely provide us with beneficial therapeutic compounds in the future.

The key role of the BMPR2 pathway and more specifically of sotatercept in PAH is reflected by the large number of new studies that are being developed to establish the role of this compound in different clinical contexts and under different forms of administration. The ZENITH clinical trial (NCT04896008) is currently investigating the role of sotatercept as rescue therapy on top of maximum tolerated background treatment in high-risk PAH patients in WHO functional class III or IV [126]. Furthermore, the HYPERION trial (NCT04811092) is assessing the effects of sotatercept in newly diagnosed (within 6 months) PAH patients at intermediate- or high-risk of disease progression [127]. Finally, the possibility of oral instead of subcutaneous administration of this compound (named KER-012) is also been evaluated in a phase-2 randomized, double-blind, placebo-controlled clinical trial [128].

The current research focused on compounds targeting pulmonary vascular remodeling sets a new need for the development of diagnostic and follow-up imaging techniques capable of directly assessing and quantifying pulmonary vascular remodeling, since current tools (electrocardiogram, echocardiography, cardiac magnetic resonance, non-invasive cardiopulmonary exercise testing, invasive rest and exercise hemodynamic evaluation) solely address it in an indirect fashion, and direct histologic visualization poses a prohibitive risk for these patients.

We find ourselves in an exciting era for PAH, where major advances in research and a shift in the therapeutic paradigm is currently being developed. Hopefully, this will lead to enrich our therapeutic options for our patients with the aim of improving their quality of life and ultimately increasing their survival.

## Figures and Tables

**Figure 1 ijms-24-04147-f001:**
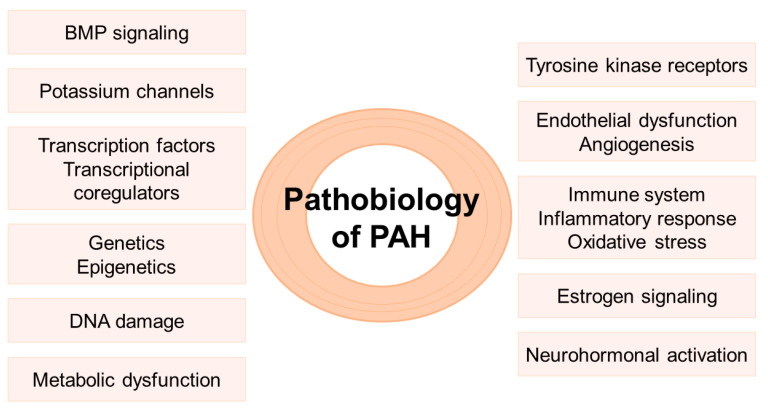
Main molecular mechanisms involved in the pathobiology of pulmonary arterial hypertension. BMP: bone morphogenic protein; PAH: pulmonary arterial hypertension.

**Figure 2 ijms-24-04147-f002:**
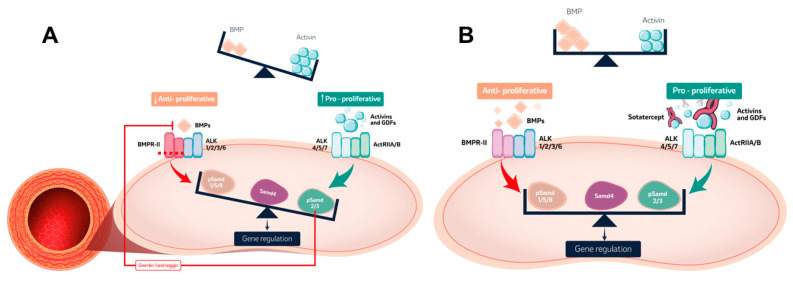
BMP signaling pathway. (**A**). Dysregulation of the BMP signaling pathway produces an imbalance between anti-proliferative and pro-proliferative routes in pulmonary arterial hypertension. Decreased BMPR2 activity leads to reduced BMP9/10-mediated signaling, diminished activation of Smad1/5/8 and reduction in the transcription of anti-proliferative genes, causing an increase in apoptosis and loss of the endothelial barrier integrity. This promotes the upregulation of pro-proliferative routes by production of activin A and GDF that bind to the ALK—ActRIIA/B complex and activate the pro-proliferative Smad2/3 pathway. This route also increases the expression of the BMP antagonists, gremlin-1 and noggin, further downregulating the BMPR2 pathway. (**B**). Sotatercept is a fusion protein comprising the extracellular domain of the activin receptor type IIA attached to the Fc domain of human IgG1. It acts as a ligand trap by binding activins and GDF and thereby restoring the balance between pro-proliferative and anti-proliferative BMP pathways. ActRIIA/B: activin receptor type IIA; ALK: activin receptor-like kinase; BMP: bone morphogenic protein; BMPR2: bone morphogenic protein receptor type-2; GDF: growth differentiation factor; pSmad: phosphorylated Smad; Smad: small mothers against decapentaplegic protein.

**Figure 3 ijms-24-04147-f003:**
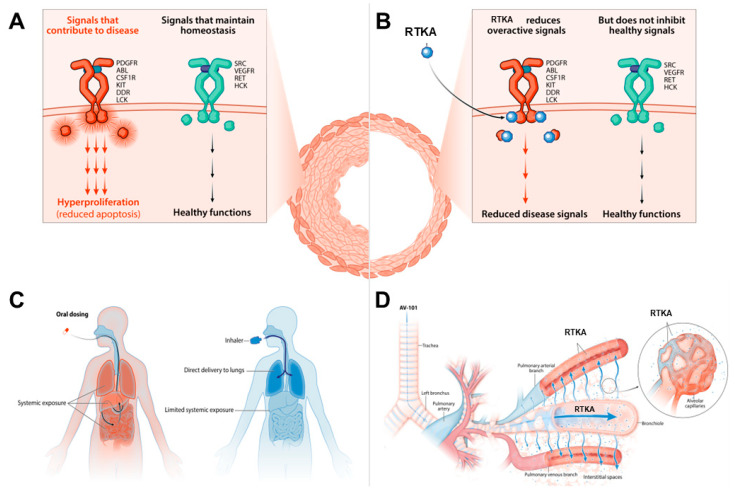
Tyrosine kinase receptor signaling. (**A**). Upregulation of the tyrosine kinase receptor PDGFR and its ligand, platelet-derived growth factor, which is the most potent mitogenic for vascular smooth muscle cells, has been observed in pulmonary arterial hypertension. Ligand binding of platelet-derived growth factor induces dimerization of the receptors followed by activation by autophosphorylation and downstream signaling and transcription of pro-proliferative genes. This leads to excessive proliferation of vascular smooth muscle cells in the arteriolar media and myofibroblasts in the vessel lumen, with media hypertrophy and intimal fibrosis. CSFIR and c-KIT are two additional PDGFR-related kinases that have also been involved in the pathobiology of PAH. CSF1R is expressed on monocytes and macrophages, which secrete platelet-derived growth factor ligands and inflammatory cytokines, promoting inflammation and remodeling of the pulmonary vasculature. C-KIT is expressed on endothelial progenitor cells and mast cells, and it is thought to play a role in perivascular inflammation and vascular remodeling. (**B**). RTKA reduce the upregulation of the pro-proliferative tyrosine kinase receptor-dependent signaling pathways. The RTKA imatinib inhibits PDGFR, DDR, c-KIT, CSF1R and ABL. The RTKA seralutinib inhibits PDGFR, CSF1R, and c-KIT, and increases bone morphogenetic protein receptor type-2. (**C**). Oral (left) and inhaled (right) administration of RTKA. (**D**). The inhaled formulation of the RTKA delivers high concentrations of the compound throughout the airways that reaches the pulmonary vasculature and the surrounding tissues to exert their anti-proliferative effects. ABL: abelson tyrosine kinase; CSF1R: colony stimulating factor 1 receptor; DDR: discoidin domain receptor tyrosine kinase; HCK: hematopoietic cell kinase; LCK: lymphocyte-specific protein tyrosine kinase; PDGFR: platelet-derived growth factor receptor; RET: rearranged during transfection tyrosine kinase; RTKA: receptor tyrosine kinase antagonist; SRC: sarcoma tyrosine kinase; VEGFR: vascular endothelial growth factor receptor.

## Data Availability

No new data were created or analyzed in this study. Data sharing is not applicable to this article.

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
