# Peer review of "Novel Molecular Mechanisms Involved in the Medical Treatment of Pulmonary Arterial Hypertension"

_ijms, 2023, doi:10.3390/ijms24044147_

Round 1

Reviewer 1 Report

Miguel and colleagues have submitted a review article titled “Novel Molecular Mechanisms Involved in the Medical Treatment of Pulmonary Arterial Hypertension.” This paper goes in detail about current findings regarding the multiple molecular mechanisms involved in the pathobiology of PAH and new molecules being developed and tested- tailored to rectify the underlying molecular mechanisms. This paper once published will be an exquisite addition compiling the recent advancements in PAH. I would suggest a minor spelling correction toward the end of the second paragraph in section 2.1 -  “……….Additionally, PASMC migration prompts neomuscularization of previously non-muscularized arterioles, and in situ thrombosis results from excessive…….” to …………resulting from excessing platelet activation in contact with a dysfunctional endothelium.

Reviewer 2 Report

Work under the direction of Dr Pilar Escribano-Subias entitled "Novel Molecular Mechanisms Involved in the Medical Treatment of Pulmonary Arterial Hypertension" was prepared and written in a thoughtful and systematic way.

The content of the text corresponds to the title and the selection of literature (123 items) is adequate and, what is important, most of the cited items come from the last 4-5 years.

The authors did not avoid minor errors that should be corrected before the publication of the article, which I personally and sincerely recommend.

1) In the introduction, the authors provide the mechanisms leading to right ventricular failure, but omit how the left ventricle behaves in this disease entity. Prolonged low preload and LV compression over time cause diastolic failure even after successful lung transplantation. There are scientific papers which, based on biopsies, indicate that the left ventricle working at a low preload in the course of PAH and having a good systolic function unfortunately has a diastolic dysfunction resulting from its fibrosis, which has been shown in hisopathological preparations. Please respond to this in the text of your work.

2) No explanation of "c-KIT" on page 9

3) There is a serious factual error on page 14 indicating that Selexipag is a prostacyclin receptor antagonist when the exact opposite is true - please be sure to correct this error. Selexipag is an agonist of prostacyclin receptor!!!

4) On page 25 in parentheses there seems to be an unfinished thought - please correct it accordingly

5) Finally, the paper lacks a discussion, which has been replaced by a chapter "Future directions" - please include a short discussion.
